# Electrically tunable layer-hybridized trions in doped WSe$_2$ bilayers

Raul Perea-Causin [1,2] ✉, Samuel Brem [3], Fabian Buchner[4], Yao Lu[5], Kenji Watanabe [6], Takashi Taniguchi [7], John M. Lupton[4], Kai-Qiang Lin [4,5] ✉ & Ermin Malic [3] ✉

Doped van der Waals heterostructures host layer-hybridized trions, i.e. charged excitons with layer-delocalized constituents holding promise for highly controllable optoelectronics. Combining a microscopic theory with photoluminescence (PL) experiments, we demonstrate the electrical tunability of the trion energy landscape in naturally stacked WSe$_2$ bilayers. We show that an out-of-plane electric field modifies the energetic ordering of the lowest lying trion states, which consist of layer-hybridized Λ-point electrons and layer-localized K-point holes. At small fields, intralayer-like trions yield distinct PL signatures in opposite doping regimes characterized by weak Stark shifts in both cases. Above a doping-asymmetric critical field, interlayer-like species are energetically favored and produce PL peaks with a pronounced Stark red-shift and a counter-intuitively large intensity arising from efficient phonon-assisted recombination. Our work presents an important step forward in the microscopic understanding of layer-hybridized trions in van der Waals heterostructures and paves the way towards optoelectronic applications based on electrically controllable atomically-thin semiconductors.

Optical and transport properties of atomically thin semiconductors are governed by tightly bound electron-hole complexes[1–4]. In the neutral regime, a weak or moderate photoexcitation generates excitons[5,6]–bound electron-hole pairs–while in doped materials each of these quasi-particles binds to an additional charge forming trions (charged excitons)[7–10]. Remarkably, the properties of excitons and trions can be tailored by stacking monolayer semiconductors into van der Waals structures[11–14]. In particular, naturally stacked WSe$_2$ bilayers host excitons that are hybrids of intra- and interlayer exciton species[15–19], combining the oscillator strength of the former[20] with the permanent dipole moment of the latter[21–23]. Moreover, the degree of layer hybridization and thus the dipole moment can be tuned by an external out-of-plane electric field[23–28]. The electrical tunability of layer-hybridized electron-hole complexes thus offers a platform to tailor

interactions, photoluminescence (PL), and transport properties of van der Waals heterostructures.

While the physics of layer-hybridized excitons is rather well established, the influence of doping has not been thoroughly understood yet. Initial efforts have focused on resolving the optical polarization of PL peaks[29–32], tuning exciton PL resonances with a gate voltage[23], calculating interlayer trion binding energies[33], and exploring the realization of Feshbach resonances[34]. Most of these studies were performed directly on twisted bilayers, whose complex properties are determined by the moiré superlattice consisting of regions with different atomic stacking[35] and involving atomic reconstruction[36]. The electrical tunability of trions in naturally stacked bilayers has been approached just recently, however, focusing only on high-energy states involving the higher-lying conduction band[37].

[1]Department of Physics, Chalmers University of Technology, Gothenburg, Sweden. [2]Department of Physics, Stockholm University, Stockholm, Sweden. [3]Department of Physics, Philipps-Universität Marburg, Marburg, Germany. [4]Department of Physics, University of Regensburg, Regensburg, Germany. [5]State Key Laboratory of Physical Chemistry of Solid Surfaces, College of Chemistry and Chemical Engineering, Xiamen University, Xiamen, China. [6]Research Center for Electronic and Optical Materials, National Institute for Materials Science, Tsukuba, Japan. [7]Research Center for Materials Nanoarchitectonics, National Institute for Materials Science, Tsukuba, Japan. ✉e-mail: causin@chalmers.se; kaiqiang.lin@xmu.edu.cn; ermin.malic@uni-marburg.de

In this theory-experiment collaboration, we provide a microscopic picture of the trion landscape in naturally stacked doped WSe$_2$ bilayers and demonstrate the electrical tunability of the trion ground state. Based on a variational solution of the trion Schrödinger equation, we find the most energetically favorable trion states to be formed by layer-hybridized electrons at the $\Lambda^{(\prime)}$ point (also denoted as Q or $\Sigma$ point in the literature[16,38,39]) of the Brillouin zone and layer-localized holes at the K$^{(\prime)}$ point, see Fig. 1. By combining experimental PL measurements with a microscopic many-body description of trion recombination, we reveal the tunability of the PL spectra by an external out-of-plane electric field. The latter serves as a tuning knob to control the layer configuration of the trion ground state (Fig. 1a, b), favoring interlayer-like trions at large electric fields. Through comparison between experiment and theory, we identify the origin of the PL signatures of p- and n-type trions, which are particularly distinct at low fields due to intricate differences in the energetic landscape of the two. In contrast, in both doping regimes, the PL at large fields is governed by interlayer-like trions, yielding a pronounced Stark red-shift of PL resonances and intense peaks owing to the efficient recombination assisted by M-point phonons (Fig. 1c). These fundamental insights serve as a guide for future studies aiming to exploit the electrical tunability of doped bilayer semiconductors.

## Results

### Layer-hybridized trion states

We consider a system of interacting electrons and holes that can tunnel between two layers in a multi-valley band structure with band-edge energies, effective masses, and tunneling strengths based on material-specific ab initio calculations[38,40]. In the regime of equally low photoexcitation and doping, the system is well described by an effective trion Hamiltonian[41] $H_{t,0}$ (see Supplementary Information, SI, for details). The eigenstates of $H_{t,0}$ are hybrid trions with the energy $E^t_{\nu \mathbf{Q}} = E^t_{\nu 0} + \hbar^2 \mathbf{Q}^2/2M_\nu$, where $M_\nu$ is the hybrid trion effective mass, $\nu = \{\nu_h, \nu_{e1}, \nu_{e2}\}$ the spin-valley configuration (for negatively-charged trions) and $\mathbf{Q}$ the center-of-mass momentum. The trion energy $E^t_{\nu 0}$ and the wave function $\Psi_{\nu l}(\mathbf{r}_1, \mathbf{r}_2)$, with the relative electron-hole coordinates $\mathbf{r}_1, \mathbf{r}_2$ and the compound layer index $l = \{l_h, l_{e1}, l_{e2}\}$, are obtained by solving the trion Schrödinger equation[42–44] generalized to describe layer hybridization,

$$\sum_{l'} \mathcal{H}_{\nu l l'}(\mathbf{r}_1, \mathbf{r}_2)\Psi_{\nu l'}(\mathbf{r}_1, \mathbf{r}_2) = E^t_{\nu 0}\Psi_{\nu l}(\mathbf{r}_1, \mathbf{r}_2). \qquad (1)$$

The Hamiltonian $\mathcal{H}_{\nu l l'}(\mathbf{r}_1, \mathbf{r}_2)$ contains the single-particle band-edge energies and kinetic terms, the Coulomb interaction between the trion's constituent particles, and interlayer tunneling processes (see Methods). The potential $V_{l_i, l_j}(\mathbf{r})$ for intra- $(l_i = l_j)$ and interlayer $(l_i \neq l_j)$ interactions is modeled following the generalization of the Rytova-Keldysh potential[45,46] to bilayer systems[47] with dielectric constants from refs. [48,49]. We focus on the qualitative description of the trion landscape and therefore disregard the impact of exchange interactions resulting in the small splitting of degenerate states[44,50]. The Schrödinger equation for hybrid trions, Eq. (1), is solved via a variational approach considering the wave function ansatz[41]

$$\Psi_{\nu l}(\mathbf{r}_1, \mathbf{r}_2) = \frac{w_l}{\mathcal{N}_l}\left(e^{-\frac{|\mathbf{r}_1|}{a_{1,l}} - \frac{|\mathbf{r}_2|}{a_{2,l}}} + C_l e^{-\frac{|\mathbf{r}_1|}{b_{1,l}} - \frac{|\mathbf{r}_2|}{b_{2,l}}}\right), \qquad (2)$$

with the variational parameters $w_l, a_{1/2,l}, b_{1/2,l}, C_l$ and the normalization factor $\mathcal{N}_l$ depending implicitly on the spin-valley index $\nu$. This wave function allows to consider an imbalance in the effective mass of the two electrons $(m_{l_{e1}} \neq m_{l_{e2}})$ or in the electron-hole interaction $(V_{l_h, l_{e1}} \neq V_{l_h, l_{e2}})$, resulting in the reduction of the trion binding energy[41].

The general theory outlined above is applied here to naturally stacked WSe$_2$ bilayers, which provide an ideal platform to exploit the

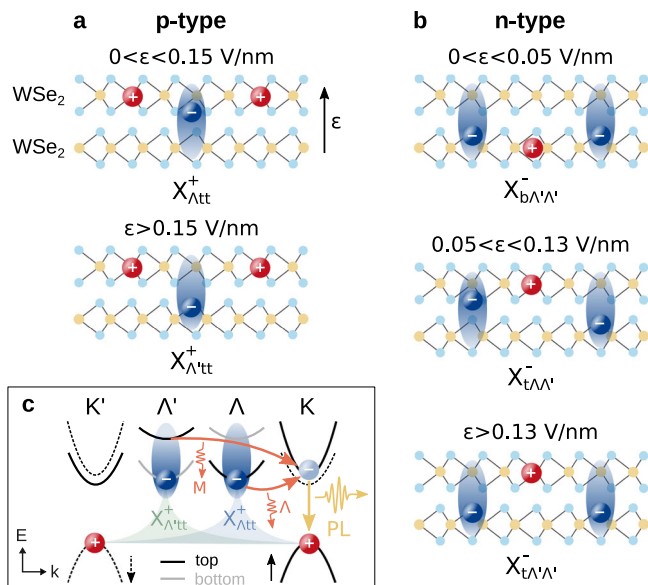

**Fig. 1 | Electrical control of the trion ground state. a, b** Layer configuration of the trion ground state in p- and n-doped WSe$_2$ bilayer for different ranges of the out-of-plane electric field $\varepsilon$. The subindices in X$^+_{\nu_e l_{h1} l_{h2}}$ (X$^-_{l_h \nu_{e1} \nu_{e2}}$) denote the electron valley $\nu_{e(1,2)}$ and hole layer $l_{h(1,2)}$. The blue ellipse surrounding the electrons illustrates the layer hybridization. **c** Reduced band structure of the WSe$_2$ bilayer. Spin-up (-down) bands from the top/bottom layer are denoted by black/gray solid (dashed) lines. Distinct PL signatures from X$^+_{\Lambda tt}$ (X$^+_{\Lambda' tt}$) trions arise from the scattering with $\Lambda$ (M) phonons to the virtual bright state followed by radiative recombination.

electrical tunability of hybrid electron-hole complexes[23,25,31,37]. The band structure in this system is characterized by valence band maxima with a large spin-orbit splitting at the K$^{(\prime)}$ point of the Brillouin zone and conduction band minima with a small and large spin-orbit splitting at the K$^{(\prime)}$ and $\Lambda^{(\prime)}$ points, respectively[38] (Fig. 1c). The ordering of the spin-split bands is reversed in opposite layers due to their 180° relative orientation. Importantly, the efficient $\Lambda$-point tunneling couples electron states in opposite layers with the same spin, whereas K-point tunneling is suppressed at the conduction band and weak at the valence band[40]. Exploiting these properties of the band structure, we introduce a concise notation to identify different trion states. The holes are located in the upper valence band at the K$^{(\prime)}$ point either in the top or bottom layer, hence their only relevant degree of freedom is the layer index ($l_h$ = t, b, i.e. top or bottom). For electrons, the valley index ($\nu_e = \Lambda, \Lambda', K, K'$) is sufficient: $\Lambda(\Lambda')$-point electrons are layer-hybridized and live mostly in the top (bottom) layer, while for K(K')-point electrons we consider those in the upper conduction band at the top (bottom) layer (Fig. 1). K$^{(\prime)}$-point electrons in the lower conduction band form high-energy dark trions which we have found to be irrelevant in PL. We note that additional degenerate states with opposite spin-valley configuration have also been taken into account but are omitted from the discussion for simplicity. With these considerations, the valley and layer configuration of p(n)-type trions is denoted in the subindex of X$^+_{\nu_e l_{h1} l_{h2}}$ (X$^-_{l_h \nu_{e1} \nu_{e2}}$).

We now solve the trion Schrödinger equation, Eq. (1), for an hBN-encapsulated WSe$_2$ bilayer and obtain the trion energy landscape shown in Fig. 2a–b (and further summarized in Table S1 in the SI). In analogy to excitons[16,23,25,40], the strong $\Lambda^{(\prime)}$-point tunneling results in a large hybridization-induced red-shift of trions with $\Lambda^{(\prime)}$ electrons[51], making these states the energetically lowest ones by far (cf. the arrow in Fig. 2a). The lowest p-type trion, X$^+_{\Lambda tt}$, contains two top-layer holes and a $\Lambda$ electron that is mostly (82%) in the top layer (Fig. 2c). A few meV above lies X$^+_{\Lambda tb}$, with one hole in the bottom layer, whereas X$^+_{\Lambda' tt}$, with a $\Lambda'$ electron that is mostly (65%) in the bottom layer, is 50 meV higher. The states X$^+_{\Lambda tt}$, X$^+_{\Lambda tb}$, and X$^+_{\Lambda' tt}$ have degenerate partners with

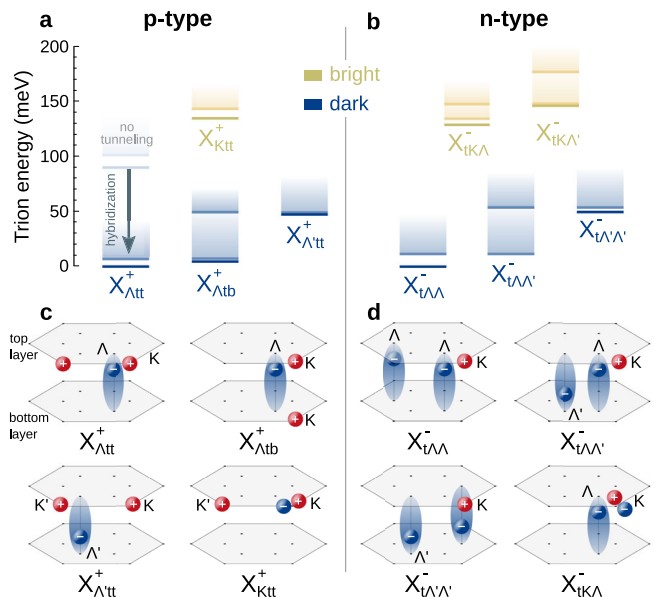

**Fig. 2 | Trion landscape. a, b** Energetic trion landscape for p- and n-type trions in bilayer WSe$_2$ (relative to the lowest state). The valley and layer configuration of p(n)-type trion states is described in the subindex of $X^+_{\nu_e l_{h1} l_{h2}}$ ($X^-_{l_{h} \nu_{e1} \nu_{e2}}$). Dark (bright) bound trion states are denoted with blue (orange) lines, while the corresponding exciton–electron continua are shaded in light-blue (orange). The large tunneling-induced energy shift at the $\Lambda^{(\prime)}$ point (denoted by an arrow in a) makes trions containing $\Lambda$ electrons the energetically lowest states. **c, d** Schematic illustration of the valley and layer configuration of different trion species.

opposite layer configuration, namely $X^+_{\Lambda' bb}$, $X^+_{\Lambda' tb}$, and $X^+_{\Lambda bb}$, respectively. For n-type trions, the lowest state, $X^-_{t\Lambda\Lambda}$, is formed by a top-layer hole and two $\Lambda$ electrons that are mostly in the top layer (cf. Figure 2b, d). $X^-_{t\Lambda\Lambda'}$ lies 10 meV above and is formed by $\Lambda$ and $\Lambda'$ electrons that are mostly in the top and bottom layer, respectively, while $X^-_{t\Lambda'\Lambda'}$, where the two electrons are mostly in the bottom layer, is 50 meV higher than $X^-_{t\Lambda\Lambda}$. The respective degenerate states with opposite layer configuration are $X^-_{b\Lambda'\Lambda'}$, $X^-_{b\Lambda\Lambda'}$, and $X^-_{b\Lambda\Lambda}$. Importantly, all these low-lying trion species are momentum-dark, i.e. they cannot directly recombine due to the momentum mismatch between K$^{(\prime)}$ holes and $\Lambda^{(\prime)}$ electrons. Bright states, with an electron and a hole located in bands with the same valley, spin, and layer indices, lie 130-140 meV above the lowest state (orange shaded lines in Fig. 2a, b), and are therefore expected to have a marginal contribution to the PL spectra at thermal equilibrium even at moderate temperatures.

In Fig. 2a–b we also show the onset energies for different exciton-electron continua (blue- and orange-shaded), which are obtained by minimizing the energy of a non-interacting exciton-electron compound with a variational hydrogenic wave function[42] generalized to bilayer systems (see SI for details). The energetic offset between the trion and the lowest exciton-electron state determines the trion binding energy. Importantly, an imbalance in the effective mass of the equal charges or in the two possible electron-hole interactions (e.g. when the two equal charges are in different layers) results in the preferential binding of one electron-hole pair and the weaker binding of the remaining charge, overall reducing the trion stability[41]. Accordingly, we observe that states with a mass and interaction balance (and where, in addition, all particles are mostly in the same layer, e.g. $X^+_{\Lambda tt}$, $X^+_{Ktt}$, $X^-_{t\Lambda\Lambda}$) have binding energies of up to 12 meV, while imbalanced states ($X^+_{\Lambda tb}$, $X^-_{t\Lambda\Lambda'}$, $X^-_{tK\Lambda}$, $X^-_{tK\Lambda'}$) have lower binding energies or are even unbound as is the case for $X^-_{t\Lambda'\Lambda'}$. Our variational approach captures the lower stability[52] or even unbound nature[33] of states where the two equal charges are in separate layers. However, since the energies obtained from the variational approach represent an upper bound, we generally underestimate the quantitative value for trion binding

energies[21,53–55], and whether $X^-_{t\Lambda\Lambda'}$ is unbound remains an open question.

The rich trion landscape described above has direct implications for the temperature dependence of PL spectra. To investigate this, we derive a PL formula for layer-hybridized trions based on a many-body equation-of-motion approach[41,56] considering the direct recombination of bright trions via electron recoil[57,58], the phonon-assisted recombination of dark trions, and phonon-induced spectral broadening[15,59] (see Methods and SI for details). The trion-phonon interaction is modeled in a spin-conserving deformation potential approach considering effective acoustic and optical modes with input parameters from ab initio calculations[60]. We evaluate and plot the PL spectra in Fig. 3 considering only the energetically lowest bound trion state for each spin-valley configuration $\nu$ that is accessed in our variational approach. The impact of the exciton-electron continuum is expected to become relevant at marginal doping values and high temperatures[61].

At 5 K, the PL spectrum is dominated by two peaks (yellow areas in Fig. 3) appearing 160-180 meV below the intralayer bright exciton resonance ($X_0$), closely resembling experimental observations[23,62]. The two peaks arise from the recombination of the energetically lowest trion $X^+_{\Lambda tt}$ ($X^-_{t\Lambda\Lambda}$) via the transition into the virtual bright state $X^+_{Ktt}$ ($X^-_{tK\Lambda}$) that is assisted by $\Lambda$ phonons with energies close to 13 meV and 30 meV[60] (Fig. 1c). The different electron-phonon coupling strength of specific phonon modes results in the unequal intensities of the two PL peaks. In particular, the higher peak involves acoustic phonons, which have a larger coupling than the optical phonons that are responsible for the low-energy peak. When the temperature is increased, higher-lying states become thermally occupied and give rise to new spectral features. At 150 K, the PL spectra for p-type doping have a large (small) contribution from $X^+_{\Lambda tb}$ ($X^+_{\Lambda' tt}$), while for n-type doping the $X^-_{t\Lambda'\Lambda'}$ trion leads to sizable signatures between -80 meV and -140 meV (orange areas in Fig. 3). Additional contributions from $X^-_{t\Lambda\Lambda'}$ could be expected but are not captured by our model as we only consider the PL arising from bound trions. Finally, bright trion PL signatures only become visible above 150 K (red areas in Fig. 3) due to their large energetic offset with respect to the trion ground state (cf. Fig. 2a, b), and dominate the spectra at room temperature. Note that in some experiments, significant PL signatures from bright states are observed even at cryogenic temperatures due to the non-equilibrium distribution created during continuous-wave high-energy optical excitation[23,25,29,62].

## Electrical control of trion photoluminescence

After having determined the valley configuration and layer mixing of the lowest lying trions as well as their PL signatures, we investigate how these states can be controlled by an out-of-plane electric field. We show a direct comparison between theoretically predicted and experimentally measured field-dependent PL.

First, we fabricated a dual-gate natural WSe$_2$ bilayer device to control both the doping and the out-of-plane electric field in the WSe$_2$ layers[37] (see Fig. S1 in the SI). We demonstrate the doping control of the energetically lowest trion states by measuring the doping dependence of the PL from the WSe$_2$ bilayer at 5 K (Fig. S2). After confirming the doping density needed to form trions, we measure the dependence of the trion PL signatures on the out-of-plane electric field while keeping a constant doping density. The out-of-plane electric field is determined from the applied bottom (top) gate voltage $V_{bg(tg)}$ via $\varepsilon = \epsilon_{hBN}(V_{bg} - V_{tg})/(\epsilon_{WSe_2}(d_{bg} + d_{tg}))$, where $d_{bg(tg)}$ is the thickness of the bottom (top) hBN insulator layers and $\epsilon_{hBN} \approx 3.4$ and $\epsilon_{WSe_2} \approx 7.2$ are the out-of-plane dielectric constants of hBN[63,64] and WSe$_2$[65]. In Fig. 4a–b we show the PL spectra for p- and n-type trions, which were measured at $V_{tg} + V_{bg} = -3$ and 0 V, respectively.

Similar to the low-temperature theoretical predictions in Fig. 3, the PL in the absence of an electric field is dominated by two peaks located 140-160 meV below the intralayer bright exciton resonance

(Fig. 4a, b). For p-type doping, as the electric field is switched on, the two peaks undergo a red-shift due to the Stark effect (Fig. 4a), reflecting the dipole length $d$ of the recombining electron-hole pair, which we extract to be $d \approx 0.14$–$0.15$ nm (see Fig. S4). In addition, a faint blue-shifting peak emerges at -135 meV. At larger fields above 0.09 V/nm, the PL becomes dominated by two intense peaks exhibiting a more pronounced red-shift (corresponding to $d \approx 0.39$ nm). In the case of n-type doping and a small electric field, the two initial peaks split into branches exhibiting opposite Stark shifts (Fig. 4b). In this regime, we can only unambiguously extract the dipole length $d \approx -0.13$ nm for the higher energy peak. At intermediate fields between 0.03 V/nm and 0.08 V/nm, the PL shows several signatures exhibiting distinct Stark shifts. In analogy to the p-type case, at larger fields the PL is dominated by two intense peaks with a more pronounced red-shift ($d \approx 0.42$ nm). For negative electric fields, we observe the same behavior for trion species with the opposite layer configuration (i.e. reversed dipole moment), see Fig. S5. We note that the field tunability of the trion PL is qualitatively similar for different doping densities in the range of $\sim 10^{11}$–$10^{12}$ cm$^{-2}$ (Fig. S5). We also emphasize that the doping densities for two layers can differ depending on the out-of-plane electric field, which however is not expected to change the main trends of the Stark shifts[37]. While our observations are similar to the previously observed field-induced switching between low- and high-dipole regimes for excitons[23,25,27,28], here we demonstrate the analogous effect for trions and revealed the asymmetric tunability of opposite doping regimes.

Now, we make use of our microscopic model to understand the underlying nature of these experimental observations. We obtain the trion energies and wave functions by solving Eq. (1) for different values of the electric field $\varepsilon$, which is incorporated into the theory via the shift of the single-particle band-edge energies. This allows us to compute the field-dependent PL spectra shown in Fig. 4c, d. Furthermore, we extract the slope of the field-induced shift of trion PL resonances, which corresponds to the dipole moment of the recombining electron-hole pair and is hence a two-body quantity (details in the SI). We emphasize that, despite trions being three-body objects, trion PL involves the recombination of one electron with one hole—therefore, the Stark shift of the PL resonance is well described by the aforementioned dipole moment. In the following, we consider a trion (lattice) temperature of 30 (4) K to simulate the experimentally realistic scenario where a high-energy continuous excitation results in a hot non-equilibrium trion distribution[58,59].

In the case of p-type trions, our calculations show that $X_{\Lambda tt}^{+}$ becomes the energetically lowest state when an electric field is applied (Fig. 4e), giving rise to the two red-shifting peaks in PL with $d = 0.13$ nm (Fig. 4c). While $X_{\Lambda tt}^{+}$ is intralayer-like (i.e. all charges are mostly located in the same layer, cf. Fig. 1a), the finite probability that electrons and holes are in opposite layers results in the small but sizable dipole moment and Stark shift. The faint blue-shifting signatures (with $d = -0.11$ nm) that appear at small fields arise from the recombination of the $\Lambda'$ electron with the bottom-layer hole in $X_{\Lambda' tb}^{+}$, which lies energetically close to the trion ground state and thus has a sizable thermal occupation. In this case, the dipole formed by the recombining electron-hole pair points in the direction opposite to the electric field leading to the observed blue-shift. In contrast, the recombination with the top-layer hole corresponds to a dipole ($d = 0.53$ nm) aligned with the field, resulting in a red-shifting signature that is hidden under the dominating $X_{\Lambda tt}^{+}$ peaks in Fig. 4c but slightly visible in the experiment (Fig. 4a). At $\varepsilon > 0.15$ V/nm, $X_{\Lambda' tt}^{+}$ becomes energetically favorable (Fig. 4e) as the electron-hole separation in the direction of the electric field is maximized (Fig. 1a), and dominates the PL with two intense peaks displaying a strong red-shift ($d = 0.50$ nm). The counterintuitively larger PL intensity of $X_{\Lambda' tt}^{+}$ despite its interlayer character is a direct consequence of the strong coupling of M-point phonons that

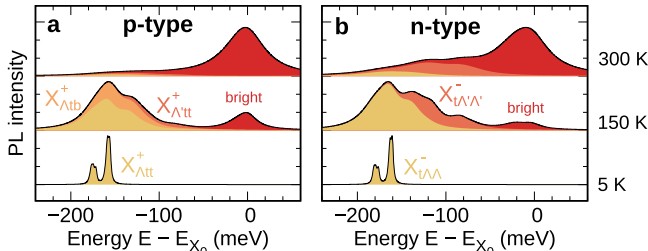

**Fig. 3 | Temperature-dependent trion photoluminescence. a** Calculated trion PL spectra in p- and (**b**) n-type bilayer WSe$_2$ at 5 K, 150 K, and 300 K with disentangled contributions from different trion states. The energy is offset with respect to the intralayer bright exciton ($X_O$) resonance, and the intensity is normalized with respect to the maximum for each temperature. Note that the two peaks in a and b at 5 K correspond to phonon sidebands arising from the same state ($X_{\Lambda tt}^{+}$ for p-type and $X_{t\Lambda\Lambda}^{-}$ for n-type doping).

govern the $\Lambda' \rightarrow K$ transition, whereas the PL arising from $X_{\Lambda tt}^{+}$ is instead governed by $\Lambda$-point phonons (Fig. 1c) characterized by weaker deformation potentials[60]. The theoretically predicted field-dependent PL (Fig. 4c) is in excellent qualitative agreement with the measured PL (Fig. 4a).

The electrical tunability of the ground state for n-type trions is characterized by the three distinct regimes sketched in Fig. 1b. First, at small electric fields, $X_{b\Lambda'\Lambda'}^{-}$ becomes the trion ground state (Fig. 4f), while $X_{t\Lambda\Lambda}^{-}$ is less favorable as the energy of $\Lambda$ electrons that are mostly in the top layer is pushed up by the electric field. Nevertheless, the two states remain energetically close to each other and are similarly populated at the trion temperature of 30 K considered, resulting in the splitting of the initial PL signatures into blue- and red-shifting peaks with $d = -0.11, + 0.13$ nm corresponding to $X_{b\Lambda'\Lambda'}^{-}$ and $X_{t\Lambda\Lambda}^{-}$, respectively (Fig. 4d). The blue- (red-)shift of the $X_{b\Lambda'\Lambda'}^{-}$ ($X_{t\Lambda\Lambda}^{-}$) signatures reflects the dipole orientation of the recombining electron-hole pair opposite to (aligned with) the electric field. Interestingly, at intermediate fields between 0.05 and 0.13 V/nm the unbound $X_{t\Lambda\Lambda'}^{-}$ becomes the lowest state. Since our theory only captures PL arising from bound trion states, we have replaced this regime by a white box in Fig. 4d. Nevertheless, we expect the presence of peaks arising from the recombination of the top-layer hole with $\Lambda$ ($\Lambda'$) electrons displaying red-shifts with $d \approx 0.1$ (0.5) nm, similar to the experimental observation.

For $\varepsilon > 0.13$ V/nm, $X_{t\Lambda'\Lambda'}^{-}$ becomes the ground state and dominates the PL with intense peaks displaying a pronounced red-shift characterized by the large dipole length $d = 0.49$ nm. The critical electric field at which the trion with larger interlayer character (i.e. $X_{t\Lambda'\Lambda'}^{-}$ and $X_{\Lambda' tt}^{+}$) becomes the ground state is thus smaller for n-type than for p-type doping. The reason for this is the smaller energy separation between the competing $X_{t\Lambda\Lambda'}^{-}$ and $X_{t\Lambda'\Lambda'}^{-}$ states (compared to that between $X_{\Lambda tt}^{+}$ and $X_{\Lambda' tt}^{+}$) that must be overcome by the field-induced shift (Fig. 4e-f). In addition, the PL intensity ratio between intra- and interlayer-like trions (at small and high fields, respectively) is larger for n-type trions (Fig. 4c-d) due to the internal structure of the virtual bright states involved in the process. Specifically, the recombining electron-hole pair in $X_{tK\Lambda'}^{-}$ (the virtual state for the recombination of $X_{t\Lambda'\Lambda'}^{-}$) is more tightly bound than in $X_{bK'\Lambda'}^{-}$ (the virtual state for the recombination of $X_{b\Lambda'\Lambda'}^{-}$), resulting in a larger trion-photon matrix element—which scales with the probability that the recombining electron and hole are at the same position (see SI for more details).

Overall, our theoretical model reproduces the main experimental observation, i.e. the ground state transition from intralayer-like to interlayer-like trion species at sufficiently large electric fields. In particular, the extracted dipole moments from the Stark shift of PL resonances in the experiment are in good agreement with the theoretical predictions (cf. Table 1). In addition, we capture the asymmetric

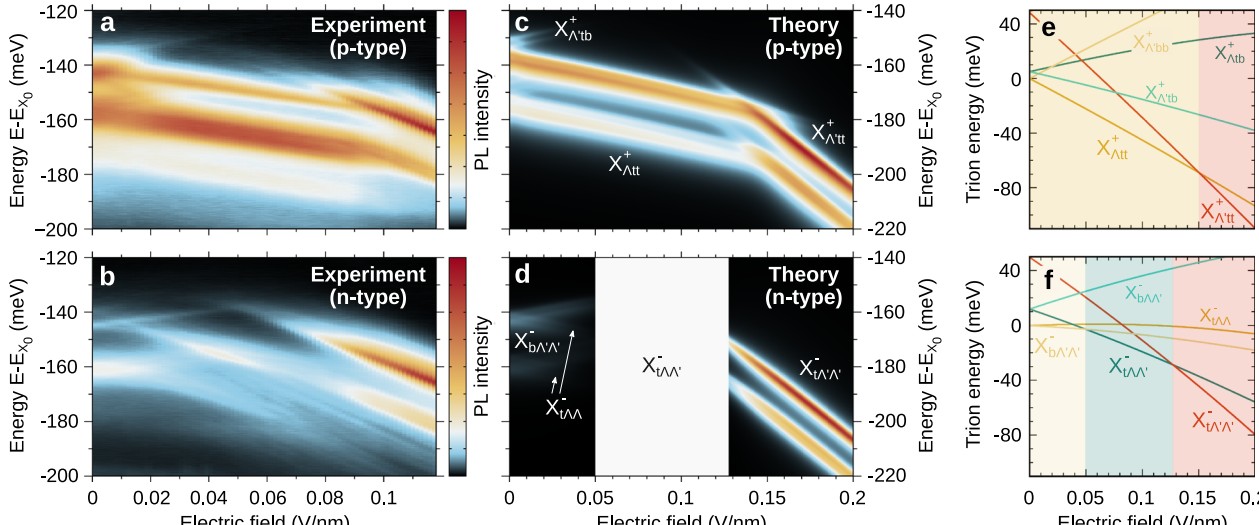

**Fig. 4 | Direct theory-experiment comparison of the electrically tunable trion photoluminescence. a, b** Experimentally measured and (**c, d**) theoretically predicted electric-field tunability of the trion photoluminescence in p- and n-type bilayer WSe$_2$. The energies are offset with respect to the intralayer bright exciton X$_0$. The trion species responsible for different peaks are denoted in (**c**) and (**d**). The white box in d corresponds to the regime governed by the unbound X$^-_{t\Lambda\Lambda'}$ state which is not captured by our PL model. **e, f** Theoretically predicted electric-field dependence of the trion energies for X$^+_{\Lambda tt}$/X$^-_{t\Lambda\Lambda}$ (dark yellow), X$^+_{\Lambda'bb}$/X$^-_{b\Lambda'\Lambda'}$ (yellow), X$^+_{\Lambda'tt}$/X$^-_{t\Lambda'\Lambda'}$ (red), X$^+_{\Lambda tb}$/X$^-_{t\Lambda\Lambda'}$ (green), and X$^+_{\Lambda'tb}$/X$^-_{b\Lambda\Lambda}$ (turquoise). The shaded areas are colored according to the energetically lowest trion state at different electric fields.

behavior of p- and n-type trion PL, including the initial peak splitting, larger intensity ratio, and lower critical field for n-type doping. The quantitative discrepancies between theory and experiment with respect to the critical electric field values for the different regimes could arise from uncertainties in the spin-orbit splitting of the Λ valley, additional screening effects from the gates, or the experimental estimation of the electric field strength.

## Discussion

We have provided a microscopic description of the electrically tunable trion energy landscape in naturally stacked WSe$_2$ bilayers. We have shown that the lowest lying trions are composed of layer-hybridized electrons at the Λ and Λ' points and layer-localized holes at the K and K' points, and dominate the PL spectra via phonon-assisted recombination. By combining our microscopic theory with experimental measurements we have demonstrated the doping-asymmetric tunability of the trion ground state via an electric field, which is manifested in distinct PL signatures. At low fields, the PL for p-type doping is governed by intralayer-like X$^+_{\Lambda tt}$ trions giving rise to two red-shifting peaks. In contrast, for n-type doping both intralayer-like X$^-_{b\Lambda'\Lambda'}$ and X$^-_{t\Lambda\Lambda}$ states are similarly populated resulting in blue- and red-shifting peaks due to their opposite layer configuration. Furthermore, we have shown that

above a doping-asymmetric critical field the PL is dominated by interlayer-like trions, resulting in peaks with pronounced red-shifts and high intensity reflecting the large electron-hole separation and the efficient scattering with M-point phonons, respectively.

The electrical tunability of the trion luminescence energy and PL intensity demonstrated here opens up new possibilities for integrating atomically thin semiconductors in optoelectronic devices where trions act as charge carriers[21]. The control over the trion layer configuration should significantly influence trion-trion interactions with important implications for charge transport[25] and for the stabilization of exotic quantum phases[66,67], and should also be relevant for the study of exciton-electron Bose-Fermi mixtures[67,68]. Our insights will guide future studies exploring and utilizing the electrical tunability of the many-body landscape of electron-hole complexes in van der Waals heterostructures.

## Methods
### Trion eigenstates
Layer-hybridized trion eigenstates fulfill the Schrödinger equation (1), which is determined by the three-body Hamiltonian

$$\mathcal{H}_{ll'}(\mathbf{r}_1, \mathbf{r}_2) = \left[ \mathcal{H}^{(0)}_l(\mathbf{r}_1, \mathbf{r}_2) + \mathcal{H}^{(C)}_l(\mathbf{r}_1, \mathbf{r}_2) \right] \delta_{ll'} + \mathcal{H}^{(tun)}_{ll'}.$$

The latter is derived by starting from the electron-hole picture (see details in SI) and it implicitly depends on the compound valley index $v$. Here, the kinetic term reads $\mathcal{H}^{(0)}_l(\mathbf{r}_1, \mathbf{r}_2) = \tilde{E}^t_l - \frac{\hbar^2 \nabla^2_{\mathbf{r}_1}}{2\mu_{l_h l_{e1}}} - \frac{\hbar^2 \nabla^2_{\mathbf{r}_2}}{2\mu_{l_h l_{e2}}} - \frac{\hbar^2 \nabla_{\mathbf{r}_1} \cdot \nabla_{\mathbf{r}_2}}{\tilde{m}_{l_h}}$, where $\tilde{E}^t_l$ is the sum of the band-edge energies of the trion's constituents, $\tilde{m}_{l_h}$ is the effective hole mass, and $\mu^{-1}_{l_h l_n}$ is the reduced electron-hole mass with $l_n$ being the single-particle layer index. The external out-of-plane electric field $\varepsilon$ is incorporated into the model via the shift of the single-particle energies, $\tilde{E}^{e/h}_{l_n} \to \tilde{E}^{e/h}_{l_n} \pm \sigma_{l_n} e_0 d\varepsilon / 2$, with + (−) for electrons (holes), $\sigma_{top} = +1$, $\sigma_{bottom} = -1$, and the layer separation $d = 0.65$ nm[48]. The Coulomb interaction between the trion's constituent particles is described by $\mathcal{H}^{(C)}_l(\mathbf{r}_1, \mathbf{r}_2) = V_{l_{e1} l_{e2}}(\mathbf{r}_2 - \mathbf{r}_1) - V_{l_h l_{e1}}(\mathbf{r}_1) - V_{l_h l_{e2}}(\mathbf{r}_2)$, where the potential for intra ($V_{l=l'}$) and interlayer

---

**Table 1 | Experimentally extracted ($d_{exp}$) and theoretically predicted ($d_{th}$) dipole lengths of the recombining electron-hole pair in the most relevant trion species**

| Trion | $d_{exp}$ (nm) | $d_{th}$ (nm) |
|---|---|---|
| X$^+_{\Lambda tt}$ | 0.14 – 0.15 | 0.13 |
| X$^+_{\Lambda'tb}$ | – | −0.11, 0.53 |
| X$^+_{\Lambda'tt}$ | 0.39 | 0.50 |
| X$^-_{b\Lambda'\Lambda'}$ | −0.13 | −0.11 |
| X$^-_{t\Lambda\Lambda}$ | – | 0.13 |
| X$^-_{t\Lambda'\Lambda'}$ | 0.42 | 0.49 |

For X$^+_{\Lambda'tb}$ two values are given corresponding to the recombination of the electron with the bottom- and top-layer hole, respectively. The exact fitting values and errors for $d_{exp}$ are shown in the SI.

$(V_{l \neq l'})$ interactions is modeled by generalizing the Rytova-Keldysh potential[45,46] to bilayer systems[47] with dielectric constants from refs. [48,49]. Finally, tunneling of the trion's constituents from one TMD layer to the other is modeled by

$$\mathcal{H}_{ll'}^{(\text{tun})} = t^e \delta_{\bar{l}_{e1}, l'_{e1}} \delta_{l_{e2}, l'_{e2}} \delta_{l_h, l'_h} + t^e \delta_{l_{e1}, l'_{e1}} \delta_{\bar{l}_{e2}, l'_{e2}} \delta_{l_h, l'_h} + t^h \delta_{l_{e1}, l'_{e1}} \delta_{l_{e2}, l'_{e2}} \delta_{\bar{l}_h, l'_h},$$

where $t^{e/h}$ is the tunneling strength for electrons/holes and $\bar{l}_n$ denotes the layer opposite to $l_n$. The energetically lowest trion state for each valley configuration is obtained by minimizing the energy $E_{\nu 0}^t$ with the wave function ansatz in Eq. (2) via a combination of global and local optimization algorithms (see SI for more details).

## Microscopic model of trion luminescence

The photoluminescence arising from layer-hybridized trions is described by extending the direct and phonon-assisted trion PL formula from ref. [41] to bilayers with finite tunneling. In particular, we consider interacting electrons, holes, phonons, and photons in a semiconducting bilayer, transform the Hamiltonian to trion basis taking only into account the Fock subspace of single trions[59], and obtain a formula for the frequency-dependent PL intensity $I_{\text{PL}}(\omega)$ (Eq. S2.5 in the SI) by exploiting the Heisenberg equation within the cluster expansion and truncation scheme[56,69]. We obtain a contribution describing the direct recombination of bright trions via the electron recoil effect[57,58], and a second contribution accounting for phonon-assisted recombination by which dark trions can emit light. More details can be found in the SI.

## Device fabrication

All materials were mechanically exfoliated from bulk crystals (WSe$_2$ from HQ Graphene, hBN from NIMS, graphite from NGS Trading and Consulting) onto silicon wafers with a 285 nm silicon dioxide top layer. At a temperature of 120 °C, a thin polycarbonate film was used to sequentially pick up a cover hBN layer, a few-layer graphene (the top gate), a top hBN layer, a WSe$_2$ bilayer, a bottom hBN layer, and a second few-layer graphene (the bottom gate). The top and bottom hBN layers have the same thickness of 32 nm, which was determined by the atomic force microscopy. This assembly was then released at a higher temperature of 170 °C on a SiO$_2$/Si substrate with pre-patterned platinum electrodes. The thin polycarbonate film was dissolved away using chloroform as a solvent and the device was further cleaned with isopropyl alcohol. The schematic diagram and the microscopic image of the dual-gate device are shown in Supplementary Fig. S1.

## Optical spectroscopy

The device was placed in a helium-flow microscope cryostat (CRYO-VAC, Konti model) and cooled down to 5 K. We used a microscope objective with a numerical aperture of 0.6 (Olympus, LUCPLFLN, 40x magnification) to focus the laser at 488 nm (Sapphire 488, Coherent GmbH) onto the sample and to collect the reflected signals. A 488 nm long-pass filter was inserted into the detection beam path to remove the laser line. The photoluminescence (PL) signal was dispersed by a diffraction grating of 1200 grooves per millimeter and detected by a charge-coupled device (CCD) camera (Princeton Instruments, PIXIS 100). The bottom and top gates of the device were connected to two separate source-measure units (Keithley 2400), and the WSe$_2$ bilayer was connected to the common ground. Supplementary Fig. S2 shows the PL spectrum of natural bilayer WSe$_2$ at different gate voltage combinations for doping dependence and electric-field dependence. Supplementary Fig. S3 shows the PL spectrum of the sample at $V_{tg} = V_{bg} = -0.3$ V, i.e. in the undoped regime.

## Data availability

All the data generated in this study are available in the article and supplementary information or from the corresponding author upon request.

## Code availability

The codes used to generate and analyse the data are available from the corresponding author upon request.

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

## Acknowledgements

R.P.-C. acknowledges fruitful discussions with Joakim Hagel and Daniel Erkensten (Chalmers University of Technology). This work has been funded by the Deutsche Forschungsgemeinschaft (DFG) via SFB 1083 and the regular project 542873285. K.-Q.L. acknowledges the Fundamental Research Funds for the Central Universities (20720230009), and funding support from the DFG via SFB 1277 (B11 314695032) and SPP 2244 (443378379). K.W. and T.T. acknowledge support from the JSPS KAKENHI (Grant Numbers 20H00354, 21H05233 and 23H02052) and World Premier International Research Center Initiative (WPI), MEXT, Japan.

## Author contributions

R.P.-C., S.B., and E.M. conceived the research and developed the theoretical model. F.B., Y.L., J.M.L., and K.-Q.L. carried out the experiments. K.W. and T.T. synthesized the hBN crystals. R.P.-C. performed the calculations and wrote the manuscript with input from all authors.

## Funding

## Competing interests

The authors declare no competing interests.
