## [Transparent Peer Review file · Nature Communications]

Electrically tunable layer-hybridized trions in doped WSe₂ bilayers

Corresponding Author: Dr Raul Perea-Causin

Version 0:

Reviewer comments:

Reviewer #1

(Remarks to the Author)

This manuscript reports layer-hybridized trions in WSe₂ bilayers that can be largely tuned by static Stark effect. The work first started with a theoretical model of specific Hamiltonians, and calculated the trion energy landscape with different configurations. With detailed analysis of these trion configurations, the manuscript assigned the PL peaks of trions in the experiments. In a dual gated device, specific electrical field dependences of these PL peaks corroborate the assignments of trion configurations, and in turn demonstrate the electrical tunability of these trions. The results are interesting and could improve the microscopic understanding of these trions in WSe₂ bilayers. However, I cannot recommend the publication of this work as I have several confusions as following.

- 1) Does the Λ point correspond to the Q point in the Brillouin zone of hexagonal lattices? As discussed in previous works, the transitions of K-Q points are usually considered as the likely ones for indirect excitons in some TMD bilayer. If so, I would suggest use the label of Q (the more frequent one) instead of Λ .
- 2) What are the origins of the two peaks in Fig. 3 at 5 K. The manuscript mentioned the lowest available trion state for the p-type is the X_+_{Λ} in Fig. 2, but labeled the second lowest peak as X_+_{Λ} . What is the origin of the lowest one? Similar scenario also applied to the X_-_{Λ} in the n-type. In the Fig. 4, these two peaks in either p-type or n-type show similar dependences of Stark effect, indicating similar dipole configuration for these two peaks. The manuscript only mentioned the second lowest one as specific interlayer trions, but did not clarify the lowest one.
- 3) There are no data about how the spectra are evolving when the WSe₂ device is tuned from the p-type to the n-type. For different doping levels, the trion PL spectra could vary quite a lot. How did the authors decide which p-type (or n-type) doping level can show these phenomena? Do these phenomena have any dependence on the doping level? Or are they only available for very specific doping condition?
- 4) The doping for the bilayer system could be very complex as well. For the dual gated devices, the doping of each layer of bilayer samples is usually analyzed separately. For instance, the doping could be applied only to the upper or the lower layer; it could also be applied evenly to the bilayers or unevenly to the bilayers. The microscopic model for these doping seems to only consider the even doping of the bilayers. But the experiment has no data to show these two layers are evenly doped.
- 5) It is really difficult to understand why the intralayer trions could show sizable Stark effect. As reported in quite some previous works, the Stark effect is negligible for intralayer excitons. How come the intralayer trions could have some considerable dipole length?
- 6) How did the authors determine the dipole lengths for different trions? In their theoretical model, the excessive electrons (holes) could be regarded as 60-80% in some specific layer based on the distributions of wavefunctions. As quasiparticles of 3-body, how can these dipole lengths be determined?
- 7) For the Stark effect at low E-fields, the intralayer trions should have a symmetric dependence of the E-field. There are only data for the positive E-field, but no data for the negative E-field. In Figs. 4a and 4b, the spectral features around zero-field do not show the symmetric trend. This style of presentation leaves the attribution of intralayer trions for the low-field Stark effect questionable.
- 8) For the Stark effect at high E-fields, the field dependence of interlayer trions is expected to be at least asymmetric. How does the Stark effect at high negative E-field look like? Would it be able to corroborate the configuration of interlayer trions?

Reviewer #2

(Remarks to the Author)

The manuscript "Electrically tunable layer-hybridized trions in doped WSe₂ bilayers" investigates doped van der Waals heterostructures hosting layer-hybridized trions. By combining theory and photoluminescence measurements on WSe₂ bilayers, they demonstrate electrical tunability of trion landscapes between intralayer-like trions and interlayer-like trions. This work represents a substantial advancement in comprehending layer-hybridized trions, thereby expanding and enhancing the realm of exciton manipulation and transport. Therefore, I suggest the publication in Nature communications after addressing minor concerns below.

1. While references 7 and 8 offer notable examples of electrically inducing charged excitons in TMD monolayers, including additional recent approaches [e.g., Nature Communications 14, 1891 (2023)] would enhance readers' comprehension and breadth of knowledge.
2. The figures should be referred in a sequential order, e.g. Fig. 1c and Fig. 1a-b.
3. The authors claim the electrical tunability of the trion lifetime. While the quantum confined Stark effect generally results in a correspondingly modified recombination rate, the authors should avoid directly mentioning control over the lifetime without presenting experimental data.
4. As studies exploring layer-hybridizations are actively ongoing, referencing pertinent literature with a specific perspective in discussion part can help guide readers toward the future direction of this field.

Reviewer #3

(Remarks to the Author)

This article addresses a long-standing challenge in the field: the modeling and characterization of interlayer hybrid excitonic species within naturally stacked bilayers. The authors employ microscopic theory to predict the intricate structure of various layer-hybridized trions and their corresponding energy landscapes within naturally stacked WSe₂ bilayers. Furthermore, they demonstrate the electrical tunability of trion energy landscape in WSe₂ bilayers by showing how an out-of-plane electric field modifies the energetic ordering of the lowest lying trion states. The results are well-presented and supported by a combination of theoretical calculations and experimental data. Overall, I believe the article merits publication. However, there are some minor issues that need addressing.

- (1) The gray box in Figure 4d could be improved for better presentation. Additionally, while the authors state that their theory only captures PL arising from bound trion states, it's puzzling why only X-t Λ ' is an unbound state, and its emission spectrum is still observed experimentally.
- (2) In the discussion section, the authors noted the 'electrical tunability of the trion luminescence energy and lifetime' but did not present the electrical tunability of interlayer-hybridized trion lifetime in the manuscript.

Author Rebuttal letter:

Response to the reviewers's comments

We thank the reviewers for carefully reading our manuscript and for providing valuable feedback which helped us to further improve the clarity of our manuscript. We provide a detailed point-by-point response to the comments below. Changes in the revised manuscript and supplementary information are marked in blue.

Reviewer: 1

1. Comment "This manuscript reports layer-hybridized trions in WSe₂ bilayers that can be largely tuned by static Stark effect. The work first started with a theoretical model of specific Hamiltonians, and calculated the trion energy landscape with different configurations. With detailed analysis of these trion configurations, the manuscript assigned the PL peaks of trions in the experiments. In a dual gated device, specific electrical field dependences of these PL peaks corroborate the assignments of trion configurations, and in turn demonstrate the electrical tunability of these trions. The results are interesting and could improve the microscopic understanding of these trions in WSe₂ bilayers. However, I cannot recommend the publication of this work as I have several confusions as following."

Answer We thank the referee for acknowledging the relevance of our results and for raising important questions, which we address below.

2. Comment "Does the $\hat{\Gamma}$ point correspond to the Q point in the Brillouin zone of hexagonal lattices? As discussed in previous works, the transitions of K-Q points are usually considered as the likely ones for indirect excitons in some TMD bilayer. If so, I would suggest use the label of Q (the more frequent one) instead of $\hat{\Gamma}$."

Answer Indeed we refer here to the Q point, also commonly referred to as Γ or Γ_2 point in the literature [16, 38, 39]. We agree that this needs to be clarified in the revised manuscript to avoid confusion.

Changes We have added in the main manuscript where we first mention the Γ point:

Γ [...] layer-hybridized electrons at the Γ point (also denoted as Q or Γ_2 point in the literature [16, 38, 39]) [...] We have decided to still keep the notation of this point as Γ to stay consistent with our and our collaborators' previous studies.

3. Comment What are the origins of the two peaks in Fig. 3 at 5 K. The manuscript mentioned the lowest available trion state for the p-type is the $X_+ \Gamma$ in Fig. 2, but labeled the second lowest peak as $X_+ \Gamma$. What is the origin of the lowest one? Similar scenario also applied to the $X_+ \Gamma$ in the n-type. In the Fig. 4, these two peaks in either p-type or n-type show similar dependences of Stark effect, indicating similar dipole configuration for these two peaks. The manuscript only mentioned the second lowest one as specific interlayer trions, but did not clarify the lowest one.

Answer The two peaks in Fig. 3 at 5 K correspond to phonon sidebands, both arising from

the same state, i.e. $X_+ \Gamma$ in the p-type case ($X_+ \Gamma$ for n-type). Since they originate from the same state, the two peaks show similar Stark effects as the referee mentions.

Note that the two peaks are both colored in the same shade as the label to illustrate that both originate from the same state indicated by the label. While the origin of the two peaks is stated in the text [The two peaks arise from the recombination of the

energetically lowest trion $X_+ \Gamma$ ($X_+ \Gamma$)], we see that the identification of the peaks in Fig. 3 should be better clarified to avoid any confusion.

1

Changes We have added in the caption of Fig. 3:

Note that the two peaks in a and b at 5 K correspond to phonon sidebands arising

from the same state (X_+

Γ for p-type and $X_+ \Gamma$ for n-type doping).

4. Comment There are no data about how the spectra are evolving when the WSe₂ device is tuned from the p-type to the n-type. For different doping levels, the trion PL spectra could vary quite a lot. How did the authors decide which p-type (or n-type) doping level can show these phenomena? Do these phenomena have any dependence on the doping level? Or are they only available for very specific doping condition?

Answer We thank the referee for bringing up the important question of how our predictions and observations depend on doping. We have verified that the electric field tunability of the PL remains qualitatively the same across doping levels on the order of 10^{11} – 10^{12} cm⁻², see Fig. R1 below. This is consistent with Fig. S2a in the SI, which shows that the spectra of negative and positive trions do not change significantly at the different doping levels explored. The change of the Fermi level only leads to a small shift of the PL resonances. To avoid the low-doping regime which can be influenced by neutral excitons, we choose a larger doping level that is nevertheless sufficiently small so that Fermi-polaron physics or trion-electron interactions are irrelevant. Thus, we choose a doping level that is representative of the single-trion physics described by our microscopic model.

[Image redacted]

Figure R1: Electric field tunability of the trion PL spectra at different n-type (top) and p-type (bottom) doping levels. The peak emerging at low doping and large fields in the bottom left panel is attributed to neutral excitons. Asymmetries in the PL spectra for large positive and negative fields arise from the fact that the device itself is not perfectly symmetric.

Changes We have added the following to the revised manuscript:

We note that the field tunability of the trion PL is qualitatively similar for different doping densities in the range of 10^{11} – 10^{12} cm⁻² (see Fig. S5).

We have included Fig. R1 in the supplementary information as Fig. S5.

2

5. Comment "The doping for the bilayer system could be very complex as well. For the dual gated devices, the doping of each layer of bilayer samples is usually analyzed separately. For instance, the doping could be applied only to the upper or the lower layer; it could also be applied evenly to the bilayers or unevenly to the bilayers. The microscopic model for these doping seems to only consider the even doping of the bilayers. But the experiment has no data to show these two layers are evenly doped."

Answer We agree that, in principle, the doping can be different in the two layers. This was in fact investigated in a recent experimental study by some of the present authors [37], where asymmetric doping led to a change in the intensity of specific trion peaks. We note that asymmetric doping is only meaningful in a many-body electron-hole system beyond the single-trion model considered here. Its description would therefore exceed the scope of this work. However, based on previous experimental observations we do not expect asymmetric doping to qualitatively alter the main observations of our study, i.e. the Stark shifts, the field-induced transition of the trion ground state from intralayer-like to interlayer-like species, and the different behaviour for p- and n-type trions.

Changes We have added the following statement to the revised manuscript:

"We also emphasize that the doping densities for two layers can differ depending on the out-of-plane electric field, which, however, is not expected to change the main trends of the Stark shifts [37]."

6. Comment "It is really difficult to understand why the intralayer trions could show sizable Stark effect. As reported in quite some previous works, the Stark effect is negligible for intralayer excitons. How come the intralayer trions could have some considerable dipole length?"

Answer We thank the referee for noting that this relevant aspect should be more clear. We agree that the Stark effect for purely intralayer excitons and trions is negligible. How-

ever, the intralayer-like trion states discussed in our work ($X_+^{\uparrow\downarrow}$ and $X_{\uparrow\downarrow}^{\uparrow\downarrow}$) are not purely intralayer but rather a hybrid of intra- and interlayer species. In particular, the recombining electron and hole have a small but non-zero vertical separation (i.e. dipole moment) as illustrated in Fig. 1a-b in the main text, resulting in a sizable Stark shift. This is consistent with previous studies reporting the Stark effect at low electric fields in the same structure for intralayer-like excitons that have a small dipole moment [23, 25, 27, 28]. Note that we denote the states where all charges have a large probability to be in the same layer as intralayer-like to distinguish them from interlayer-like states, where negative and positive charges reside mostly in different layers.

Changes We have added a sentence to the revised manuscript to better clarify this point:

"While $X_+^{\uparrow\downarrow}$ is intralayer-like (i.e. all charges are mostly located in the same layer, cf. Fig. 1a), the finite probability that electrons and holes are in opposite layers results in the small but sizable dipole moment and Stark shift."

7. Comment "How did the authors determine the dipole lengths for different trions? In their theoretical model, the excessive electrons (holes) could be regarded as 60-80% in some specific layer based on the distributions of wavefunctions. As quasiparticles of 3-body, how can these dipole lengths be determined?"

Answer This is indeed a very important point. As in the experiment, the theoretical dipole lengths are extracted from the slope of the shift of the PL resonance energies as a function of electric field. It can be shown in our microscopic approach that the trion PL resonances shift with the electric field following the Stark effect of the recombining electron-hole pair (see updated supplementary information). Thus, the dipole lengths

3

discussed throughout our work correspond to the (two-body) dipole formed by the recombining electron and hole.

Changes We have added a sentence to the revised main manuscript:

"Furthermore, we extract the slope of the field-induced shift of trion PL resonances, which corresponds to the dipole moment formed by the recombining electron-hole pair and is hence a two-body quantity (details in the SI)."

We have also extended the supplementary information to show how the Stark shift of trion PL resonances is related to the dipole moment of the recombining electron-hole pair.

8. Comment âFor the Stark effect at low E-fields, the intralayer trions should have a symmetric dependence of the E-field. There are only data for the positive E-field, but no data for the negative E-field. In Figs. 4a and 4b, the spectral features around zero-field do not show the symmetric trend. This style of presentation leaves the attribution of intralayer trions for the low-field Stark effect questionable.

For the Stark effect at high E-fields, the field dependence of interlayer trions is expected to be at least asymmetric. How does the Stark effect at high negative E-field look like? Would it be able to corroborate the configuration of interlayer trions? â

Answer We thank the referee for bringing up these important points, which we have further clarified by explicitly showing the spectra for both positive and negative fields in Fig. R1.

It is very important to be clear about what has a symmetric, antisymmetric, or asymmetric dependence on the electric field. The resonance energy of purely intralayer excitons or trions indeed evolves symmetrically since these lack a permanent dipole [Nat. Commun. 9, 1633 (2018)]. In our work, however, the energetically lowest states at small electric fields (intralayer-like trions) have a significant interlayer component and therefore a permanent dipole moment. Thus, both intralayer-like and interlayer-like trions exhibit a linear Stark effect, i.e. their resonance energies shift linearly with the electric field and hence have an antisymmetric dependence on the field. Importantly, in photoluminescence spectra at low temperatures and thermal equilibrium only the energetically lowest (i.e. most occupied) state is observed. This explains the symmetric dependence of the PL on the electric field (Fig. R1), where the dominating states at negative fields have the opposite layer configuration and dipole moment as those for positive fields. This is consistent with previous studies on excitons in the same system [23, 25, 27, 28]. In addition, we note that the Stark effect in the considered system is not expected to be asymmetric for positive and negative fields, as the homobilayer is inversion symmetric.

Changes We have included the new Fig. R1 in the supplementary information and added the following sentence in the main text:

âFor negative electric fields, we observe the same behaviour for trion species with the opposite layer configuration (i.e. reversed dipole moment), see Fig. S5.â

Reviewer: 2

1. Comment âThe manuscript âElectrically tunable layer-hybridized trions in doped WSe₂ bilayersâ investigates doped van der Waals heterostructures hosting layer-hybridized trions. By combining theory and photoluminescence measurements on WSe₂ bilayers, they demonstrate electrical tunability of trion landscapes between Intralayer-like trions and interlayer-like trions. This work represents a substantial advancement in

4

comprehending layer-hybridized trions, thereby expanding and enhancing the realm of exciton manipulation and transport. Therefore, I suggest the publication in Nature communications after addressing minor concerns below.â

Answer We are very grateful for the refereeâs clear recommendation for publication of our work in Nature Communications and for the constructive comments, which helped us to improve the presentation of our manuscript.

2. Comment âWhile references 7 and 8 offer notable examples of electrically inducing charged excitons in TMD monolayers, including additional recent approaches [e.g., Nature Communications 14, 1891 (2023)] would enhance readersâ comprehension and breadth of knowledge.â

Answer We thank the referee for the valuable suggestion and for bringing this recent work to our attention.

Changes We have added new citations including the one suggested by the referee.

3. Comment âThe figures should be referred in a sequential order, e.g. Fig. 1c and Fig. 1a-b.â
Changes We now refer to the figures in sequential order.

4. Comment âThe authors claim the electrical tunability of the trion lifetime. While the quantum confined Stark effect generally results in a correspondingly modified recombination rate, the authors should avoid directly mentioning control over the lifetime without presenting experimental data.â

Answer In principle, the modulation of the PL intensity in our calculations indicates that the trion radiative lifetime is tuned. It is in this sense that we show the tunability of the trion lifetime. However, we see that the statement will be clearer if we refer instead to the PL intensity, which is also what we measure experimentally.
Changes We have replaced âlifetimeâ by âPL intensityâ.

5. Comment âAs studies exploring layer-hybridizations are actively ongoing, referencing pertinent literature with a specific perspective in discussion part can help guide readers toward the future direction of this field.â

Changes We have extended the discussion and added pertinent citations:
âThe control over the trion layer configuration should significantly influence trion-trion interactions with important implications for charge transport [25] and for the stabilization of exotic quantum phases [66, 67], and should also be relevant for the study of exciton-electron Bose-Fermi mixtures [67, 68].â

Reviewer: 3

1. Comment âThis article addresses a long-standing challenge in the field: the modeling and characterization of interlayer hybrid excitonic species within naturally stacked bilayers. The authors employ microscopic theory to predict the intricate structure of various layer-hybridized trions and their corresponding energy landscapes within naturally stacked WSe₂ bilayers. Furthermore, they demonstrate the electrical tunability of trion energy

5

landscape in WSe₂ bilayers by showing how an out-of-plane electric field modifies the energetic ordering of the lowest lying trion states. The results are well-presented and supported by a combination of theoretical calculations and experimental data. Overall, I believe the article merits publication. However, there are some minor issues that need addressing.â

Answer We thank the referee for the positive assessment of our manuscript and for the insightful comments, which helped us improve the presentation of our work.

2. Comment âThe gray box in Figure 4d could be improved for better presentation. Additionally, while the authors state that their theory only captures PL arising from bound trion states, itâs puzzling why only X₁ is an unbound state, and its emission spectrum is still observed experimentally.â

Answer We thank the referee for the suggestion to improve the presentation of our results. Regarding the X₁ trion, we note that the wave function ansatz considered gives an upper bound for the trion energy, which in our calculations approaches the onset of the exciton-electron continuum from above, i.e. showing that the state is unbound based on the ansatz. A more sophisticated calculation could allow for more quantitative statements, but our main focus here lies on the electrical tunability between intralayer-like and interlayer-like trions, and we hope that our results will trigger further theoretical and experimental studies clarifying the still open questions. This is why we chose to mark this region in the plot in Fig. 4d as a gray box. To improve the presentation, we have changed the grey box to a white one.

Changes We have changed the grey box to white and added the sentence:
âHowever, since the energies obtained from the variational approach represent an upper bound, we generally underestimate the quantitative value for trion binding energies [21, 52-54], and whether X₁ is unbound remains an open question.â

3. Comment In the discussion section, the authors noted the electrical tunability of the trion luminescence energy and lifetime but did not present the electrical tunability of interlayer-hybridized trion lifetime in the manuscript.

Answer In principle, the modulation of the PL intensity in our calculations indicates that the trion radiative lifetime is tuned. It is in this sense that we show the tunability of the trion lifetime. However, we see that the statement will be clearer if we refer instead to the PL intensity, which is also what we measure experimentally.

Changes We have replaced lifetime by PL intensity.

6

Version 1:

Reviewer comments:

Reviewer #1

(Remarks to the Author)

The authors have made many efforts to address my concerns, and I am glad to see the manuscript gets improved. Before I can recommend its publication, there are still two confusions to be clarified.

1. The authors claimed the two peaks in Fig. 3 are phonon sidebands of X^+_{Att} (similar to X^-_{Att}). Can the authors clarify why the higher energy peak shows higher intensity than the lower one? What kind of electron-phonon coupling may be included here? There are not too many references about the phonon sidebands of trions, and most of them are theoretical predictions without experimental verification. The labeling of phonon sidebands is critical for the claim of this work. It is strongly suggested to avoid handwaving without experimental clarification on specific electron-phonon coupling.

2. In the Stark effect characterization of these trions, I am not sure if the two-body dipole moment can realistically represent these various species of trions. Though the trions may show linear Stark effect as reported in this work, the two-body dipole moment may not be accurate to explain these phenomena. The PL of trions involves recombination of three particles, and their intervalley (or intravalley) scattering depending on specific three-body configurations. Instead of over-simplified two-body dipole moment model, the Stark effect is strongly suggested to be elaborated with the realistic three-body recombinations.

Reviewer #2

(Remarks to the Author)

I believe the revised manuscript is ready for publication.

Reviewer #3

(Remarks to the Author)

The authors have adequately addressed my concerns. I recommend acceptance of this manuscript for publication.

Author Rebuttal letter:

Response to the reviewer's comments

1. Comment The authors have made many efforts to address my concerns, and I am glad to see the manuscript gets improved. Before I can recommend its publication, there are still two confusions to be clarified.

Answer We are thankful for the referee's constructive comments, which indeed helped us to improve the manuscript. We are happy to address the two additional comments below. Changes in the revised manuscript are marked in blue.

2. Comment The authors claimed the two peaks in Fig. 3 are phonon sidebands of X^+_{Att} (similar to X^-_{Att}). Can the authors clarify why the higher energy peak shows higher intensity than the lower one? What kind of electron-phonon coupling may be included here? There are not too many references about the phonon sidebands of trions, and most of them are theoretical predictions without experimental verification. The labeling of phonon sidebands is critical for the claim of this work. It is strongly suggested to avoid handwaving without experimental clarification on specific electron-phonon coupling.

Answer The referee addresses an important point. The PL shown in Fig. 3 corresponds to theoretical calculations, while the experimental measurements are shown in Fig. 4a-b. The assignment of the two peaks in Fig. 3 as phonon sidebands follows directly from their microscopic origin, namely the phonon-assisted recombination term in the PL formula (the second term between brackets in Eq. S2.5). The larger intensity of the high-energy peak is due to the stronger trion-phonon coupling of that specific recombination channel. In particular, the higher peak involves $\hat{\Gamma}$ -point acoustic phonons, which have a larger electron-phonon coupling than the optical $\hat{\Gamma}$ -point phonons that are responsible for the low-energy peak. The coupling is theoretically described via a deformation potential approach with potential parameters extracted from ab initio calculations in Ref. 60, where different interaction mechanisms were considered. The assignment of the experimental peaks (Fig.4a-b) as phonon sidebands arising from

$X+\hat{\Gamma}$ or $X\hat{\Gamma}$ is based on the fact that their energy and electric field dependence are in very good agreement with our theoretical predictions (Fig.4c-d). Note that we have benchmarked our microscopic theory in multiple other joint theory-experiment studies including the prediction of exciton phonon-sidebands [Nano Lett. 2020, 20, 4, 2849-2856; ACS Photonics 2020, 7, 10, 2756-2764] as well as the prediction of phonon-assisted charge transfer and relaxation dynamics [Nature 608, 499-503 (2022); Sci. Adv. 10, eadi1323 (2024); Nano Lett. 2024, 24, 15, 4505-4511].

Changes We have specified in the caption of Fig. 3 that the data shown corresponds to calculations. We have also added in the main text: "The different electron-phonon coupling strength of specific phonon modes results in the unequal intensities of the two PL peaks. In particular, the higher peak involves acoustic phonons, which have a larger coupling than the optical phonons that are responsible for the low-energy peak."

3. Comment "In the Stark effect characterization of these trions, I am not sure if the two-body dipole moment can realistically represent these various species of trions. Though the trions may show linear Stark effect as reported in this work, the two-body dipole moment may not be accurate to explain these phenomena. The PL of trions involves recombination of three particles, and their intervalley (or intravalley) scattering depending on specific three-body configurations. Instead of over-simplified two-body dipole moment model, the Stark effect is strongly suggested to be elaborated with the realistic three-body recombinations."

1

Answer We agree with the referee that characterizing trion recombination with a two-body quantity seems counter-intuitive, but the crucial point here is that only one electron and one hole recombine, leaving behind an additional electron or hole. Therefore, the PL of trions involves the recombination of two particles and not three. Thus, the (two-body) dipolar description of the PL Stark shift is not a simplification, but it is actually backed up by our full theory of trion recombination. In particular, based on our microscopic model, we show in the SI (see "Trion Stark effect and dipole moment") that the Stark shift of trion PL resonances corresponds to the Stark shift of the recombining electron-hole pair within the trion, and is therefore described by the dipole moment of the recombining electron-hole pair.

Changes We have added in the main text: "[...] corresponds to the dipole moment of the recombining electron-hole pair and is hence a two-body quantity (details in the SI). We emphasize that, despite trions being three-body objects, trion PL involves the recombination of one electron with one hole—therefore, the Stark shift of the PL resonance is well described by the aforementioned dipole moment."

2

Version 2:

Reviewer comments:

Reviewer #1

(Remarks to the Author)

The authors have addressed my concerns, and I recommend the publication of this manuscript.
